# BETTER GENERATIVE REPLAY FOR CONTINUAL FEDERATED LEARNING

**Daiqing Qi**[1]**, Handong Zhao**[2]**, Sheng Li**[1]
[1]University of Virginia, [2]Adobe Research
{daiqing.qi, shengli}@virginia.edu, hazhao@adobe.com

## ABSTRACT

Federated Learning (FL) aims to develop a centralized server that learns from distributed clients via communications without accessing the clients' local data. However, existing works mainly focus on federated learning in a single task scenario. with static data. In this paper, we introduce the continual federated learning (CFL) problem, where clients incrementally learn new tasks and history data cannot be stored due to certain reasons, such as limited storage and data retention policy [1]. Generative replay (GR) based methods are effective for continual learning without storing history data. However, we fail when trying to intuitively adapt GR models for this setting. By analyzing the behaviors of clients during training, we find the unstable training process caused by distributed training on non-IID data leads to a notable performance degradation. To address this problem, we propose our FedCIL model with two simple but effective solutions: 1. model consolidation and 2. consistency enforcement. Experimental results on multiple benchmark datasets demonstrate that our method significantly outperforms baselines. Code is available at: `https://github.com/daiqing98/FedCIL`.

## 1 INTRODUCTION

Federated learning (McMahan et al., 2017) is an emerging topic in machine learning, where a powerful global model is maintained via communications with distributed clients without access to their local data. A typical challenge in federated learning is the non-IID data distribution (Zhao et al., 2018; Zhu et al., 2021a), where the data distributions learnt by different clients are different (known as heterogeneous federated learning). Recent methods (Li et al., 2020; Chen & Chao, 2020; Zhu et al., 2021b) gain improvements in the typical federated learning setting, where the global model is learning a single task and each client is trained locally on fixed data. However, in real-world applications, it is more practical that each client is continuously learning new tasks. Traditional federated learning models fail to solve this problem.

In practice, history data are sometimes inaccessible considering privacy constraints (e.g., data protection under GDPR) or limited storage space (e.g., mobile devices with very limited space), and the unavailability of previous data often leads to catastrophic forgetting (McCloskey & Cohen, 1989) in many machine learning models. Continual learning (Thrun, 1995; Kumar & Daume III, 2012; Ruvolo & Eaton, 2013; Chu & Li, 2023) aims to develop an intelligent system that can continuously learn from new tasks without forgetting learnt knowledge in the absence of previous data. Common continual learning scenarios can be roughly divided into two scenarios (Van de Ven & Tolias, 2019): task incremental learning (TIL) and class incremental learning (CIL) (Rebuffi et al., 2017). In both scenarios, the intelligent system is required to solve all tasks so far. In TIL, task-IDs of different task are accessible, while in CIL, they are unavailable, which requires the system to infer task-IDs. The unavailability of task-IDs makes the problem significantly harder.

In this paper, we propose a challenging and realistic problem, continual federated learning. More specifically, we aim to deal with the class-incremental federated learning (CI-FL) problem. In this setting, each client is continuously learning new classes from a sequence of tasks, and the centralized server learns from the clients via communications. It is more difficult than the single FL or CIL,

---

[1]https://eur-lex.europa.eu/legal-content/EN/TXT/PDF/?uri=CELEX:32016R0679

because both the non-IID and catastrophic forgetting issues need to be addressed. Compared with traditional continual learning settings that only involve one model, our problem is more complex because there are multiple models including one server and many clients.

Owning to the success of generative replay ideas in continual learning, we propose to adopt Auxiliary Classifier GAN (ACGAN) (Odena et al., 2017), a generative adversarial network (GAN) with an auxiliary classifier, as the base model for the server and clients. Then the generator of the model can be used for generative replay to avoid forgetting. Interestingly, experiments show that a simple combination of ACGAN and FL algorithms fails to produce promising results. It is already known that the unstable training process (Roth et al., 2017; Kang et al., 2021; Wu et al., 2020) and imbalanced data can worsen the performance of generative models. We find that this phenomenon becomes more severe in the context of federated learning, which leads to the failure of the intuitively combined model.

To overcome these challenges, we propose a federated class-incremental learning (FedCIL) framework. In FedCIL, the generator of ACGAN helps alleviate catastrophic forgetting by generating synthetic data of previous distributions for replay, meanwhile, it benefits federated learning by transferring better global and local data distributions during the communications. Our model is also subject to the privacy constraints in federated learning. During the communications, only model parameters are transmitted. With the proposed global model consolidation and local consistency enforcement, our model significantly outperforms baselines on benchmark datasets.

The main contributions of this paper are as follows:

- We introduce a challenging and practical problem of continual federated learning, i.e., class-incremental federated learning (CI-FL), where a global model continuously learns from multiple clients that are incrementally learning new classes *without* memory buffers. Subsequently, we propose generative replay (GR) based methods for this challenge.

- We empirically find the unstable learning process caused by distributed training on highly non-IID data with popular federated learning algorithms can lead to a notable performance degradation of GR based models. Motivated by this observation, we further propose to solve the problem with model consolidation and consistency enforcement.

- We design new experimental settings and conduct comprehensive evacuations on benchmark datasets. The results demonstrate the effectiveness of our model.

## 2   RELATED WORK

**Class-Incremental Learning.** Class-incremental learning (CIL) (Rebuffi et al., 2017) is a hard continual learning problem due to the unavailability of the task-IDs. Existing approaches to solve the CIL problem can be divided into three categories (Ebrahimi et al., 2020), including the replay-based methods (Rolnick et al., 2019; Chaudhry et al., 2019), structure-based methods (Yoon et al., 2017), and regularization-based methods (Kirkpatrick et al., 2017; Aljundi et al., 2018). In addition, the generative replay based methods (Shin et al., 2017; Wu et al., 2018) belong to the replay-based methods, but they do not need a replay buffer for storing previous data. Instead, an additional generator is trained to capture the data distribution of data learnt so far, and it generates synthetic data for replay when the real data becomes unavailable. The generative replay based methods have obtained promising results on various CIL benchmarks, where storing data is not allowed (data-free).

**Federated Learning.** Federated learning has been extensively studied in recent years. We mainly discuss the ones that involve generative models in this section. For instance, some recent federated learning methods train a GAN across distributed resources (Zhang et al., 2021; Rasouli et al., 2020) in a federated learning paradigm. They are subject to privacy constraints in that the parameters of the GAN are shared instead of the real data. In our work, we also strictly follow the privacy constraints by only transmitting GAN parameters.

**Continual Learning and Federated Learning.** So far, only few works lie in the intersection of federated learning and continual learning. Casado et al. (2020) discussed federated learning with changing data distributions, but it only involves single-task scenario of federated learning. Yoon et al. (2021) introduced a federated continual learning setting from the perspective of continual learning. Our work is significantly different from it. First, in (Yoon et al., 2021), the federated

continual learning focuses on the performance of **each single client**, and thus it does not maintain a global model on the server side. Our objective is opposite to it in that we aim to develop **a global model** that incrementally learns from the clients. Second, the method in (Yoon et al., 2021) focuses on the **task-incremental learning**, where task-IDs are required for inference, while our work aims to deal with the more challenging task of **class-incremental learning**, where task-IDs are not available. In addition, the methods proposed in (Usmanova et al., 2021; Guo et al., 2021; Hendryx et al., 2021) are also relevant to our work, all of which have a global model that incrementally learns new classes. In particular, federated reconnaissance (Hendryx et al., 2021) uses prototype networks for its proposed task. It is different from our work in that, in federated reconnaissance, new classes maybe overlapped with old classes, which is not allowed in the standard class incremental scenario. Also, it focuses on few-shot learning and uses a buffer to store prototypes of learnt classes, which is not needed in our framework. (Guo et al., 2021) is very relevant to (Hendryx et al., 2021), which allows overlapped classes among a client's private tasks. Thus it is inconsistent with the standard incremental learning (IL) setting. (Usmanova et al., 2021) is different from us in the communication settings. Dong et al. (2022) used a memory buffer to store old data, which decreases the difficulty to a large extend. We focus on generative replay based continual learning methods which works with no memory buffers. We further clarify the differences between our work and the most related methods in Appendix D.

## 3  PRELIMINARY

In this section, we first present the problem definitions. Then we clarify the differences between our setting and existing work.

### 3.1  PROBLEM FORMULATION

**Federated Learning (FL).** A typical Federated Learning problem can be formalized by assuming a set $\boldsymbol{C} = \{\mathcal{C}_1, \mathcal{C}_2, ..., \mathcal{C}_n\}$ of $n$ different clients, where each client $\mathcal{C}_k$ owns its private data $\mathcal{D}_k$ with its corresponding task $\mathcal{T}_k$, and a global model parameterized by $\boldsymbol{\theta_g}$ is deployed on a centralized server, which communicates with the clients during their training on their own local data. The aim of FL is to learn a global model that minimizes its risk on each of the client tasks. where $\mathcal{L}_k$ is the objective of $\mathcal{T}_k$. On the other hand, we present the definition of class-incremental learning, which is a challenging continual learning problem as task-IDs are not available.

**Class-Incremental Learning (CIL).** In class-incremental learning, a model sequentially learns from a sequence of data distributions denoted as $\boldsymbol{D} = \{\mathcal{D}^1, \mathcal{D}^2, ..., \mathcal{D}^m\}$, where each $\mathcal{D}^i$ is the data distribution of the corresponding task $\mathcal{T}^i$ and label space is $\mathcal{Y}^i$. During $\mathcal{T}^t$, all data distributions $\{\mathcal{D}^i | i < t\}$ will be unavailable, and the goal of CIL is to effectively learn from $\mathcal{D}^t$, while maintaining its performance on learnt tasks. After all $m$ tasks, the model is expected to map samples from $\boldsymbol{D} = \{\mathcal{D}^1, \mathcal{D}^2, ..., \mathcal{D}^m\}$ to $\mathcal{Y}^1 \cup \mathcal{Y}^2 ... \cup \mathcal{Y}^m$ without Task-IDs.

**Class-Incremental Federated Learning (CI-FL).** In CI-FL, each client $\mathcal{C}_i$ sequentially learns from a series of $m$ tasks locally (denoted as $\boldsymbol{T}_i = \{\mathcal{T}_i^1, \mathcal{T}_i^2, ..., \mathcal{T}_i^m\}$) in a class-incremental way, during which time, the clients communicate with global model on centralized server. The goal of CI-FL is to maintain a *global model* that predicts all classes seen by all clients so far.

### 3.2  DESIDERATA

We have some desiderata for our proposed setting, which makes our class-incremental federated learning more general, realistic and challenging, compared with existing works. Discussions including differences with existing works are available in Appendix D.

## 4  METHODOLOGY

We propose a new framework, FedCIL, for class-incremental federated learning, as shown in Fig. 1. In this section, we first review ACGAN, which is adopted as the base model for server and clients. Then we discuss an interesting observation, i.e., simply combining ACGAN and federated learning algorithms fails to work well. Motivated by the findings, we finally present our FedCIL framework.

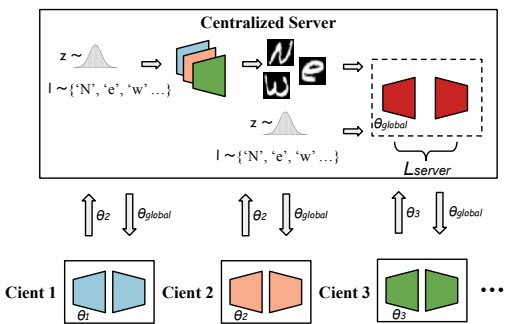 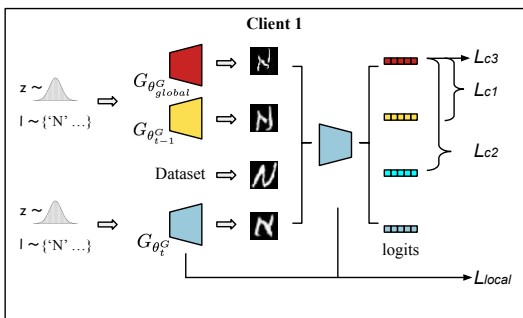

Figure 1: Framework of our FedCIL. The left figure illustrates the *model consolidation* on the server side. During each communication, the server model is first initialized from merged clients, then consolidated with synthetic samples generated by client generators. The right figure illustrates *consistency enforcement* on client side (client 1 for example), which is achieved by consistency loss functions applied to the output logits of the classification module of the client.

### 4.1 ACGAN

Generative replay based methods are popular in a strict CIL setting, where storing data is not allowed (data-free). DGR (Shin et al., 2017), a representative generative replay based method, trains a GAN for replay in addition to the classification model. ACGAN (Odena et al., 2017; Wu et al., 2018) is better than DGR in that, it reduces the complexity of the model by integrating the the classifier and the generative model as a single model. Besides, it enables conditional generation, which is crucial for learning an unbiased classifier. Thus we choose ACGAN as the base model for each client.

In ACGAN, each sample is generated by providing the noise $z$ and a class label $c$ to the generator. The objective function of the generator can be written as:

$$\mathcal{L}_{\text{gen}} = \min_{\theta^{\text{G}}} \left( \mathcal{L}_{\text{gan}}^{\text{G}}(\theta, X) + \mathcal{L}_{\text{ce}}^{\text{G}}(\theta, X) \right), \tag{1}$$

$$\mathcal{L}_{\text{gan}}^{\text{G}}(\theta, X) = -\mathbb{E}_{z \sim p_z, c \sim p_c} \left[ \log(D_{\theta^{\text{D}}} \left( G_{\theta^{\text{G}}}(z, c) \right)) \right], \tag{2}$$

$$\mathcal{L}_{\text{ce}}^{\text{G}}(\theta, X) = -\mathbb{E}_{z \sim p_z, c \sim p_c} \left[ y_c \log(C_{\theta^{\text{C}}} \left( G_{\theta^{\text{G}}}(z, c) \right)) \right], \tag{3}$$

where $C$, $D$ and $G$ are the classification module, discriminator module and the generator, respectively. Note that $C$ and $D$ share the feature extractor layers and only differ in the last out put layer. $\mathcal{L}_{\text{ce}}^{G}(\theta, X)$ and $\mathcal{L}_{\text{gan}}^{G}(\theta, X)$ are respectively the standard cross-entropy loss for classification and the generator loss in a typical GAN. $X$ is the training dataset. $\theta$ denotes the overall parameters. $\theta^{\text{C}}$, $\theta^{\text{D}}$, and $\theta^{\text{G}}$ are the parameters for classification module, discriminator module and the generator, respectively. $p_c$ is the label distribution, and $p_z$ is the noise distribution. $y_c$ is the ground truth label. The corresponding discriminator loss function is:

$$\mathcal{L}_{\text{dis}} = \min_{\theta^{\text{D}}, \theta^{\text{C}}} \left( \mathcal{L}_{\text{gan}}^{\text{D}}(\theta, X) + \mathcal{L}_{\text{ce}}^{\text{D}}(\theta, X) \right), \tag{4}$$

$$\mathcal{L}_{\text{gan}}^{\text{D}}(\theta, X) = -\mathbb{E}_{(x,c) \sim X} \left[ \log(D_{\theta^{\text{D}}}(x)) \right] - \mathbb{E}_{z \sim p_z, c \sim p_c} \left[ \log(1 - D_{\theta^{\text{D}}} \left( G_{\theta^{\text{G}}}(z, c) \right)) \right], \tag{5}$$

$$\mathcal{L}_{\text{ce}}^{\text{D}}(\theta, X) = -\mathbb{E}_{(x,c) \sim X} \left[ y_c \log(C_{\theta^{\text{C}}}(x)) \right] - \mathbb{E}_{z \sim p_z, c \sim p_c} \left[ y_c \log(C_{\theta^{\text{C}}} \left( G_{\theta^{\text{G}}}(z, c) \right)) \right]. \tag{6}$$

Overall, we write the loss function of ACGAN as:

$$\mathcal{L}_{acgan}(\theta, X) = \mathcal{L}_{\text{gen}} + \mathcal{L}_{\text{dis}}. \tag{7}$$

When trained properly, The ACGAN can generate synthetic images while it can also give predictions of real images.

### 4.2 WHY DOES INTUITIVE COMBINATION FAIL TO WORK WELL?

However, simply combining the ACGAN and a federated learning algorithm to adapt ACGAN to heterogeneous federated learning yields unpromising results. The instability of ACGAN during training, which is aggravated in the context of non-IID federated learning leads to the degradation

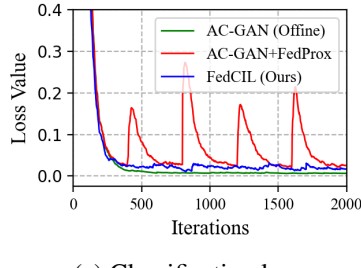

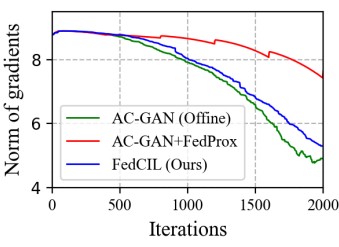

(a) Classification loss           (b) Gradient Norm

Figure 2: The classification losses and gradient norms (i.e., Eq. (6)) of (1) ACGAN + FedProx, (2) ACGAN (offline) and (3) FedCIL (Ours) of a randomly selected client during a period of training on EMNIST.

of the model performance. Fig. 2 illustrates this phenomenon. In the simply combined model, the classification loss experiences undesired peaks in each communication round, which impedes the training of the generator in ACGAN.

In a single model scenario, the gradient exploding in the classifier of ACGAN have a negative impact on the training of the generator, which appears more often when the classifier is not well-trained yet, e.g., on the early training stage. When it happens, the classification loss from Eq. (6) comes into dominance and breaks the balance between the classification loss from Eq. (6) and the discrimination loss from Eq. (5). The generation ability will be limited due to such over-tilt since then. *Finally the generation ability is impaired, which impacts the quality of generative replay, leading to more forgetting.* Therefore, we need to prevent the classification loss from growing too high at the early stages of each communication round.

The standard cross-entropy loss function is written as:

$$\mathcal{L}_{\text{ce}} = -\mathbb{E}_{x \sim X}[\log(\frac{\exp(F(x)^\top w_y)}{\sum_{j=1}^c \exp\left(F(x)^\top w_j\right)})] \tag{8}$$

where $x$ is the instance from the training data distribution $X$ and $\boldsymbol{y}$ is its label, $F$ is the feature extractor and $W = \{w_1, ..., w_c\}$ is the parameter set of the classifications layer. $c$ is the number of total classes.

Then the derivative of Eq. (8) w.r.t $w_{k \in \{1,2,...,c\}}$ is:

$$\frac{\partial \mathcal{L}_{\text{CE}}}{\partial w_k} = -\mathbb{E}_{x \sim X}[F(x)(\mathbf{1}_{y=k} - \log(\frac{\exp(F((x,y))^\top w_k)}{\sum_{j=1}^c \exp\left(F(x)^\top w_j\right)}))] \tag{9}$$

where $\mathbf{1}_{y=k}$ is the indicator function whose value equals to 1 if $y = k$ else 0.

Eq. (9) indicates that the gradient norm of $\mathcal{L}_{\text{ce}}$ is associated with the feature norm (i.e., the norm of $F(x)$) and the classification possibilities. Thus the generation ability is more likely to degrade when the classifier gives inaccurate predictions.

The way clients are trained in federated learning makes such degradation more often. In each communication round, each local client first synchronizes parameters with the global model, which worsens the performance of the classifier on its local dataset. According to Eq. (9), the gradient norm tends to grow, which can lead to the gradient exploding. And the large values of $\mathcal{L}_{\text{ce}}^{\text{D}}(\theta, X)$ from Eq. (6) break the balance between $\mathcal{L}_{\text{ce}}^{\text{D}}(\theta, X)$ and $\mathcal{L}_{\text{gan}}^{\text{D}}(\theta, X)$. The over-tilt to $\mathcal{L}_{\text{ce}}^{\text{D}}(\theta, X)$ limits the generation ability and finally decreases the model performance, which relies on the quality of generative replay. Moreover, the training process in federated learning consists of many communications rounds, which makes it worse because such degradation happens in early stages of every communication round, throughout the whole training process.

Figures 2 and 4 illustrates this process. At the beginning of each communication round, the classification loss and FID score of ACGAN experiences an obviously peak if simply combining ACGAN with a FL algorithm (FedProx is used in experiments). It is observed every 400 iterations as we set local intention $T = 400$. In a single model scenario without federated learning, i.e., ACGAN (offline), the loss is stable. Our model works in federated learning while the loss curve is comparable

to ACGAN (offline). Note that ACGAN (offline) indicates the ideal case. Because in FL, clients are synchronized with the server at every communication round and offline training is unavailable.

## 4.3 FEDCIL

To address the problems, we propose a new framework (Fig. 1) for class-incremental federated learning. We denote all parameters of a single client model $C_i$ as $\theta_i$ and that of the global model $G$ as $\theta_{global}$. Clients sequentially learn a series of non-overlapped classes. Meanwhile, the server communicates with the clients multiple rounds to learn all classes seen by all clients so far. During each round, clients first learn the current or new classes from their own dataset for certain iterations. Then the model parameters of selected clients are sent to the server, where the global model aggregates the parameters and conduct a model consolidation to enrich itself with the knowledge from different clients. Finally, the parameters of the global model are sent back the all clients.

**Model consolidation**. On the server side, the centralized server first receives the parameters (denoted by $\theta_1, \theta_2, ...$) sent by selected clients $C_{selected} = \{C_1, C_2, ...\}$. Then the parameters are merged to initialize the global model. Note that in CI-FL setting, clients continuously learn new classes, thus new output nodes are added to the global model if any of the clients learns new classes. In a practical CI-FL setting, the local data of clients are usually non-IID and more importantly, new classes are continuously collected and learnt by clients, thus simply merging the parameters often fails in such case. We proposed to solve this problem by *model consolidation*. The global model is first initialized with the merged parameters and an ensemble of classification heads from different clients. Then we generate balanced synthetic data with collected generator parameters from selected clients. Finally the global model is consolidated by the ACGAN loss function $\mathcal{L}_{acgan}$ with generated data:

$$\mathcal{L}_{server} = \mathcal{L}_{acgan}(\theta_{global}, X^g), \tag{10}$$

where $X^g$ are synthetic data generated by collected generators this communication round and $\theta_{global}$ is the parameter of the generator on the server.

**Consistency enforcement**. On the client side, each client owns three modules during the local training: (1) a ACGAN (2) a copy of the generator of itself $G_{\theta^G}^{t-1}$ (yellow generator in Fig. 1) at the end of last task (3) the global generator $G_{\theta_{global}^G}$ (red generator in Fig. 1) which it received in the last communication. In local training, the client first initializes itself with received parameter $\theta_{global}$ from the global model. During the training, the client is expected to effectively learn from new classes while maintain its knowledge on learnt classes. For this purpose, $G_{\theta^G}^{t-1}$ generates samples $X_{t-1}^g$ conditioned on labels from previous tasks and they are mixed up with training data $X$ to form the input for ACGAN loss:

$$\mathcal{L}_{local} = \mathcal{L}_{acgan}(\theta, X_{t-1}^g \cup X), \tag{11}$$

Although the classifier of a client is trained to give same predictions of the real images and the synthetic images generated by generators if they have the same label, their features can be different because their sources are different.

Denote generated samples by the generator of current model, the generator from last task, and the generator from received global model as $X_{t-1}^g$ and $X_{\mathbf{g}}^g$ and denote the real sample as $X$. Given $X_{t-1}^g$, $X_{\mathbf{g}}^g$ and $X$, to learn a better global model, their features are expected be close to each other when their labels are the same. In this case, it is easier to reach a global optimality instead of sticking in a local optimality or diverges. We propose to achieve it by *consistency enforcement*, i.e., aligning the their features produced by intermediate layers of the classifier. To make it simple, instead of aligning features output by intermediate layers, we try to align their output logits by the last layer, i.e., the classification layer, which is common in knowledge distillation. In this way, given different images with the same label but from different generators or the real dataset, the classifier tends to output similar distributions. Thus the classifier is more robust against images from different distributions. We use consistency loss functions to ensure this:

$$\mathcal{L}_{c1} = \min_{\theta^C}(-\mathbb{E}_{x_{t-1}^g \sim X_{t-1}^g, x_{\mathbf{g}}^g \sim X_{\mathbf{g}}^g}[D_{KL}(f(\theta^C, x_{t-1}^g) \parallel f(\theta^C, x_{\mathbf{g}}^g))]) \tag{12}$$

$$\mathcal{L}_{c2} = \min_{\theta^C}(-\mathbb{E}_{x \sim X, x_{\mathbf{g}}^g \sim X_{\mathbf{g}}^g}[D_{KL}(f(\theta^C, x) \parallel f(\theta^C, x_{\mathbf{g}}^g))]), \tag{13}$$

$$\mathcal{L}_{c3} = \min_{\theta^C}(-\mathbb{E}_{x_{\mathbf{g}}^g \sim x_{\mathbf{g}}^g, y(x_{\mathbf{g}}^g) \sim Y}[D_{KL}(f(\theta^C, x_{\mathbf{g}}^g) \parallel y(x_{\mathbf{g}}^g))]), \tag{14}$$

$$\mathcal{L}_{\text{c}} = \mathcal{L}_{\text{c1}} + \mathcal{L}_{\text{c2}} + \mathcal{L}_{\text{c3}}. \tag{15}$$

Finally, the overall loss function for a client is:

$$\mathcal{L}_{client} = \mathcal{L}_{local} + \mathcal{L}_c, \tag{16}$$

where $D_{KL}(P \parallel Q)$ is the Kullback–Leibler divergence, a measure of how one probability distribution Q is different from the second probability distribution P. $\theta^C$ is the parameter related to the classification module and $f(\theta^C, x)$ are the softmaxed output logits of the classification head.

Eq. (13) can also be interpreted as using some soft labels for the classification task of real samples. With proper soft label, the classification module can be trained more stably and has better generalization thus can have better performance(Phuong & Lampert, 2019). It is especially fitful for our CI-FL setting because better generalization benefits both class incremental learning (Mirzadeh et al., 2020) (Buzzega et al., 2020) and federated learning, where non-IID data requires better generalization of the model. Intuitively, Eq. (14) can transfer knowledge from the global model to the client by the generated data $X_{global}^g$ to enrich the client with global knowledge.

**Privacy.** To protect privacy in federated learning, only sharing parameters is a simple yet effective solution. Related works that train a GAN across distributed resources (Zhang et al., 2021; Rasouli et al., 2020) adopts this strategy. We also adopt this solution and mainly focus on addressing the catastrophic forgetting problem in CI-FL. Some advanced methods like Private FL-GAN (Xin et al., 2020), which specifically address the privacy issue, can be integrated into our framework if higher levels of privacy are required.

## 5 EXPERIMENTS

### 5.1 DATASETS

**Choice of Datasets.** We adopt commonly used datasets for federated learning (FL) (Zhu et al., 2021b), suggested by the LEAF benchmark (Caldas et al., 2018). Proper modifications are made according to the definition of our CI-FL setting. For vision tasks, MNIST, EMINST and CelebA are suggested. We do not use CelebA because it is built as a binary classification task in FL and not suitable for continual learning setting. Besides, we add use CIFAR-10, which is a simple but still challenging dataset in data-free class-incremental learning (CIL). They are simple datasets for classification, but difficult and far from being solved in non-IID FL and CIL (Prabhu et al., 2020). For all datasets: **MNIST**, **EMINST-Letters**, **EMNIST-Balanced** and **CIFAR-10**, we build 5 tasks for each client with 2 classes every task. Details of datasets and data processing can be found in Appendix A.

### 5.2 BASELINES

We compare our *FedCIL* approach with the following baselines, including two representative FL models and three models for continual federated learning. Details of baselines can be found in Appendix B. **FedAvg** (McMahan et al., 2017). **FedProx** (Li et al., 2020). A representative FL model which is better at tackling heterogeneity in federated networks than FedAvg. **FedLwF-2T** (Usmanova et al., 2021). It exploits the idea of Learning without Forgetting(LwF) (Li & Hoiem, 2017) in federated learning. **FedProx/FedAvg+ACGAN Replay** (Also referred as Fed-Prox/FedAvg+ACGAN in the paper). It is a combination of FedProx/FedAvg and ACGAN. **Fed-Prox/FedAvg+DGR** (Shin et al., 2017). It is a combination of FedProx/FedAvg and Deep Generative Replay (DGR) (Shin et al., 2017).

### 5.3 EXPERIMENTAL SETUP

**Setting**. We use five clients in all experiments and all of them participate in each communication with the server, following related works (Usmanova et al., 2021; Hendryx et al., 2021), where five or less than five clients are used. In (Yoon et al., 2021), the same setting is also used on non-iid-50 dataset. Taking CIL into account when dealing with FL is very challenging, thus existing works and our work adopt this setting to simplify the case for analysis.

**Configuration.** Unless otherwise specified, we set local training iteration $T = 400$ and global communication round $R = 200$ for all models. As we build five tasks for each client, there are 40

Table 1: Results on MNIST, EMNIST-Letters, EMNIST-Balanced and CIFAR-10. We report the accuracy of the global model when all clients finish their training on all tasks, which is referred as global accuracy.

| Model | MNIST | EMNIST-L | EMNIST-B | CIFAR-10 |
|---|---|---|---|---|
| FedAvg (McMahan et al., 2017) | $72.28 \pm 0.82$ | $19.36 \pm 0.95$ | $17.25 \pm 0.25$ | $27.21 \pm 2.39$ |
| FedProx (Li et al., 2020) | $72.84 \pm 0.73$ | $19.69 \pm 0.75$ | $17.74 \pm 0.55$ | $27.43 \pm 2.46$ |
| FedLwF-2T (Usmanova et al., 2021) | $75.61 \pm 0.93$ | $23.91 \pm 0.78$ | $17.22 \pm 0.90$ | $27.02 \pm 2.38$ |
| FedAvg+DGR | $97.46 \pm 0.51$ | $71.92 \pm 0.74$ | $63.55 \pm 0.46$ | $37.93 \pm 2.27$ |
| FedProx+DGR | $97.55 \pm 0.48$ | $71.83 \pm 0.65$ | $63.55 \pm 0.27$ | $37.87 \pm 2.47$ |
| FedAvg+ACGAN Replay | $97.13 \pm 0.35$ | $73.85 \pm 0.17$ | $66.87 \pm 0.79$ | $38.31 \pm 2.64$ |
| FedProx+ACGAN Replay | $97.38 \pm 0.63$ | $73.91 \pm 0.29$ | $66.19 \pm 0.92$ | $38.34 \pm 2.55$ |
| FedCIL (Ours) | $\mathbf{99.13 \pm 0.34}$ | $\mathbf{78.15 \pm 0.30}$ | $\mathbf{73.12 \pm 0.47}$ | $\mathbf{45.27 \pm 2.42}$ |

(a) FedCIL (Ours)      (b) FedProx+DGR      (c) FedProx+AGCAN

Figure 3: The confusion matrices for the global model (classifier) on the server in FedCIL, Fed-Prox+DGR and FedProx+ACGAN.

rounds per task. For each local iteration, we adopt mini-batch size $B = 32$ for MNIST, EMNIST-L, EMNIST-B and $B = 100$ for CIFAR-10. The number of generated samples in a iteration is the same as the mini-batch size. The backbone of the feature extractor is a 3-layer CNN and that of the generator is a 4-layer CNN with batch normalization layers. The Adam optimizer is used with the learning rate 1e-4. We run each experiment three times and report the mean and standard deviation.

## 5.4 RESULTS AND ANALYSIS

Table 1 shows the results on MNIST, EMNIST-Letters, EMNIST-Balanced and CIFAR-10. From the results we can see catastrophic forgetting becomes more notable as the similarity between tasks of different clients reduces. FedLwF-2T (Usmanova et al., 2021) outperforms FedAvg and FedProx on MNIST and EMNIST-Letters. But on EMNIST-Letters and EMNIST-Balanced, with overlapped classes becoming less or almost none, its performance drops significantly. FedProx/FedAvg+DGR and FedProx/FedAvg+ACGAN Replay outperforms other baselines by a large margin, which is more obvious when the setting becomes more difficult. Our model FedCIL further gains more improvement compared to the two models. We further summarize how our proposed two methods benefit continual federated learning and illustrate it with examples in Appendix E. More discussions and results on FedCIL are available in Appendix F.

**FedCIL learns better generators.** To prevent forgetting, generative replay based methods exploit a generator to generate data for replay. Thus the quality of generated images has an important influence on the model performance. The generators of clients in our model generate better data than others. As stated in the weakness of simple combination, directly training ACGAN in heterogeneous federated learning setting with a FL algorithm leads to the performance degradation of generators.

This is because the sever-client synchronization in every communication round damages each client's classification performance on its local dataset at the early training stages of each round. Such classification accuracy decrease then lead to much larger classification loss according to Eq. (9). Fig. 2 shows this phenomenon. The unexpected huge classification loss can break the balance between the losses from Eq. (5) and Eq. (6), thus model will overly tilt to the classification part. Eventually it weakens the performance of generator.

From Fig. 2 it is observed that our model doesn't suffer from this problem. Our propose model consolidation and consistency enforcement effectively prevent the gradient exploding in early stages of

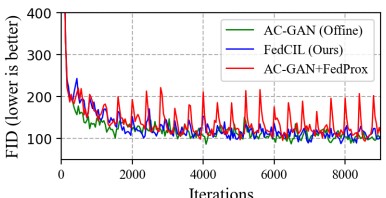

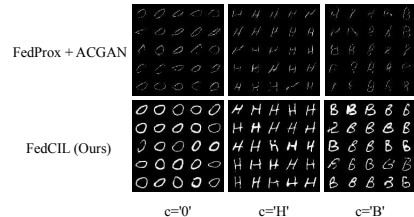

Figure 4: FID scores of ACGAN + FedProx, ACGAN (offline) and ours. A random client is selected during training on EMNIST.

Figure 5: Visualization of generated images from clients in FedProx+AC-GAN (first row) and FedCIL (second row).

every communication round, which guarantees the stability of the generator. Fig. 5 shows the generated images by FedCIL and FedProx+ACGAN. It visualizes that generators in FedCIL generates images of higher quality than FedProx+ACGAN. FID score can be a measurement of the quality of generated images (lower is better). Fig. 4 shows that generators in our model is more stable and have better FID score than FedProx+ACGAN during the training.

**FedCIL learns less biased classifiers and generators**. Fig. 3 shows the confusion matrices for the global models in FedCIL, FedProx+DGR and FedProx+ACGAN on EMNIST-Letters. It is observed that the global classifier in our model is less biased compared with two baselines. In our CI-FL setting, each client is learning $n$ classes with $x$ instances for one class during a task. If some clients are learning same classes, the proportion of this class will be larger than other classes. Thus when the server and clients are synchronized, the server model is prone to be biased towards this class. Our model consolidation on the server side can effectively alleviate this phenomenon by consolidating the merged model with balanced generated data. In this way the knowledge on the less frequent classes can be strengthened. On the client side, with Eq. (15), the global knowledge can be better transferred to the local clients, preventing the local clients from overfitting to its local dataset.

**FedCIL allows more effective server-client communication**. Different from class-incremental learning (CIL), where we only need to address the problem of catastrophic forgetting in a single model, both server and clients models need to be considered in CI-FL. From the perspective of CIL, each model can of course learn from the history state of itself. However, with other models available in the CI-FL, they are also expected to learn from each other. Compared with FedProx+DGR, where each client keeps a private GAN for generative replay and only synchronize classifiers with the server model, the generators in FedCIL can effectively learn form others for better generation ability. Thus the generative replay can be more effective and achieve better performance.

Table 2: Ablation study

| Method | Accuracy |
|---|---|
| Full method | **78.15 ± 0.30** |
| Ablate model consolidation (Eq. 10) | 75.02 ± 0.41 |
| Ablate consistency enforcement (Eq. 15) | 73.91 ± 0.29 |
| Ablate generative replay | 19.36 ± 0.95 |

**Ablation Study**. We perform ablation studies on EMNIST-Letters to evaluate the contribution of each component. Table 2 proves the effectiveness of our proposed methods. Each module can improve the model performance effectively. The results are consistent with our analysis above.

**Influence of local iterations**. We study how the local iterations $T$ affects our model in Appendix C.

## 6  CONCLUSION

In this paper, we introduce a practical and challenging federated continual learning scenario, where a centralized server continuously learns from clients that are incrementally learning new tasks from a streaming of local data. Furthermore, we find simply adapting generative replay ideas into the setting fails to work as expected. Then we analyze its weaknesses and propose a new model FedCIL to address the problems. Experimental results on benchmark datasets manifest its effectiveness.

ACKNOWLEDGEMENT

This research is supported by the the U.S. Army Research Office Award under Grant Number W911NF-21-1-0109 and Adobe Data Science Research Award.

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

# A   DATASET

- **MNIST**: It is a digit image classification dataset of 10 classes with a training set of 60,000 instances and a test set of 10,000 instances. We split instances of each class into non-overlapped parts. Each client randomly picks up two classes without replacement to form a task and it is repeated for five times for form a sequence of five classes. Note that for any client, if a class already exists in any of its tasks, the client will not pick up this class again. This ensures that for any client, a class will not appear more than once in its private sequence, which is subject to the definition of class-incremental learning. In the setting, classes from different clients are highly overlapped. Note that although it is highly overlapped from the overall view, however, during each task, clients are still learning different tasks, the data is non-IID and it is still a challenging federated learning problem.

- **EMNIST-Letters**: It is a character classification dataset of 26 classes containing 145,600 instances. We process this dataset similar to MNIST and each client also randomly picks up two classes to form a task and repeats five times in total. It is harder because classes from different clients are less overlapped and the sever eventually learns more classes.

- **EMNIST-Balanced**: It is a digit and character classification dataset of 47 classes containing 131,600 instances. We follow the same steps to process this dataset. It is the hardest because there are almost no overlapped classes from different clients and the server learns more classes.

- **CIFAR-10**: It is an image classification dataset of 10 classes with 60,000 instances. The process on this dateset is the same as MNIST.

Figure 6 illustrates the how we build tasks for clients.

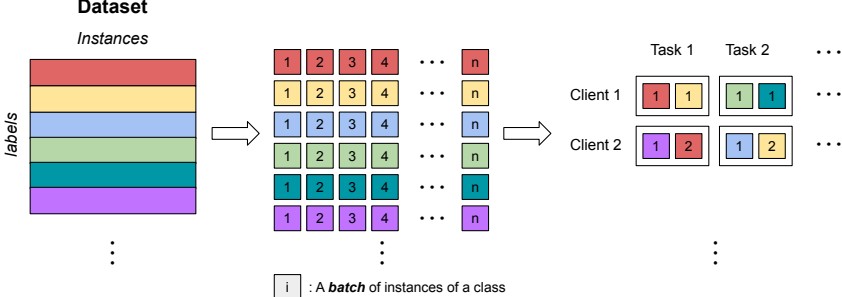

Figure 6: Illustration of how to build tasks for clients. Different color represents different classes. We split instances of each class into non-overlapped parts. Each client randomly picks up two classes without replacement to form a task and it is repeated for five times for form a sequence of five classes. Note that for **a client**, if a class already exists in any of its tasks, the client will not pick up this class again. This ensures that for any client, a class will not appear more than once in its private sequence, which is subject to the definition of class-incremental learning.

# B   BASELINES

- **FedAvg** (McMahan et al., 2017) : It is a representative federated learning model, which aggregates client parameters in each communication. It is a simply yet effective model for federated learning.

- **FedProx** (Li et al., 2020) : It is also a representative federated learning model, which is better at tackling heterogeneity in federated networks than FedAvg.

- **FedLwF-2T** (Usmanova et al., 2021). It uses the idea of Learning without Forgetting(LwF) (Li & Hoiem, 2017) in federated learning. LwF keeps a copy of itself at the end of each task, which acts as a teacher to remind the current model of previous knowledge via knowledge distillation (Hinton et al., 2015). FedLwF-2T is similar to LwF and uses the global model as the second teacher during the local training of the clients.

- **FedProx/FedAvg+ACGAN Replay** (Also referred as FedProx/FedAvg+ACGAN in the paper). It is a combination of FedProx/FedAvg and ACGAN. Each client is an ACGAN

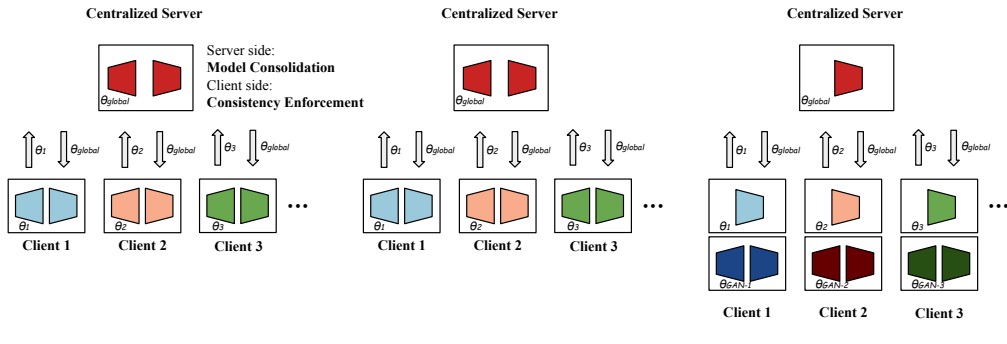

Figure 7: Illustration of our model and baselines.

based model and uses generative replay (i.e., exploiting the samples generated by the generator of the ACGAN for replay to prevent forgetting) locally. The global model and the clients are synchronized by FedProx or FedAvg.

- **FedProx/FedAvg+DGR** (Shin et al., 2017). It is a combination of FedProx/FedAvg and Deep Generative Replay (DGR) (Shin et al., 2017). Each client consists of a GAN and a classifier. The GAN is trained additionally for generative replay. The train process of a client is the same as the process in (Shin et al., 2017). The global model and the clients (the classifiers) are synchronized by FedProx or FedAvg.

Figure 7 illustrates our model and baselines.

## C    INFLUENCE OF LOCAL ITERATIONS

To study how the number of local iterations affect our model, we experiment on EMNIST-Letters with different local iterations from 50 to 1000. The accuracy curve tends to be more gentle as the number of local iteration increases, which is helpful when the local training time is limited. We can have a compromise between performance and speed according to different application scenarios.

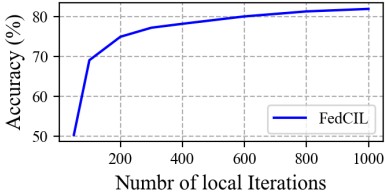

Figure 8: Influence of different local iterations

## D    DESIDERATA

We have some desiderata for the proposed setting, which makes the class-incremental federated learning more general, realistic and challenging, compared with existing works.

1. For a client, it is continuously learning new classes, and the learnt classes will be unavailable in future tasks. We strictly follow a typical continual learning setting, in which there are no overlapped classes among tasks, and thus more challenging. This is a significant difference from (Hendryx et al., 2021) and (Guo et al., 2021), where any instance only appears once but different instances of the same class may appear in different tasks in the task sequence of a client.

2. For different clients, there may be overlapped classes in their respective private tasks. This is practical because each client is learning their own tasks without communicating with each other. Thus, it is reasonable that we do not put extra assumptions on the overall data distribution of all clients.

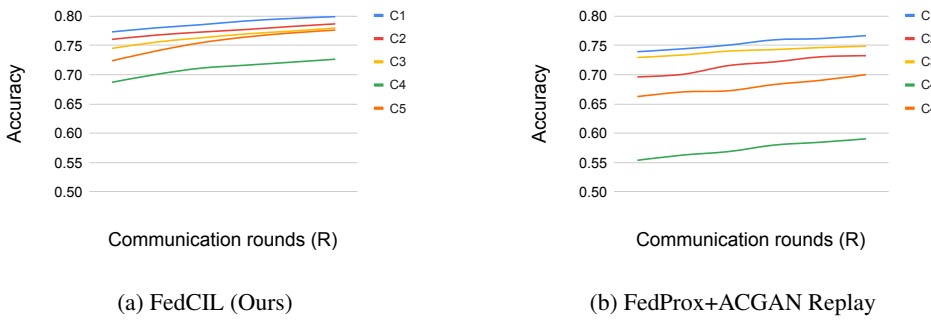

(a) FedCIL (Ours)  (b) FedProx+ACGAN Replay

Figure 9: Case study on EMNIST. We record the performance of each client (1-5) on its own local dataset **immediately** after it receives the parameters from the global model. We plot this performance of each client for several communication rounds during training.

3. The clients are not allowed to store previous data in a buffer, which is consistent with (Usmanova et al., 2021) and (Yoon et al., 2021).

4. The clients are not allowed to communicate with each other at any time. To follow the privacy constraints in typical federated learning, neither the raw data sharing between the server and the clients nor among the clients is allowed.

5. At any given time-step $t$, the global model should be able to classify all classes that all clients collected and learned so far. It means that the global model is dynamically updated all the time, which makes it a powerful lifelong federated learning model. On the contrast, (Yoon et al., 2021) learns client models and does not maintain a global model that discriminates all classes without task-IDs.

6. The number of communication rounds should be flexible according to different settings. This is different from (Usmanova et al., 2021), where a round of communication happens when a client finishes its learning on a local dataset, which will be no longer available in the next round. Our assumption is more general and follows a typical continual learning paradigm.

# E  CASE STUDY

In Section 4.2, we demonstrate when the generation ability of ACGAN degrades and how this phenomenon is aggravated in the context of federated learning. To summarize, the performance of ACGAN degrades when the gradient norm (Eq. (9)) grows too large and breaks the balance between $\mathcal{L}_{ce}^{D}(\theta, X)$ and $\mathcal{L}_{gan}^{D}(\theta, X)$. The over-tilt to $\mathcal{L}_{ce}^{D}(\theta, X)$ limits the generation ability and finally decreases the model performance.

The gradient exploding happens in early stages of **each** local training (Fig. 2 and Fig. 4 ). According to Eq. (9), the gradient norm is associated with 1. the predicted class probability distribution and 2. feature norms. Our model consolidation and consistency enforcement are designed from the two perspectives.

## E.1  PREDICTED CLASS PROBABILITY DISTRIBUTION

The gradient norms of baseline models are very large at early stage of local training because the predictions are highly inaccurate at the beginning. Received global model are directly merged from clients from last round via FedAvg or FedProx. Empirically we find such merging hurts the local performance of clients to a large extend and yields highly inaccurate predictions on local data, leading to large gradient norms (Eq. (9)).

### E.1.1  CASES: WHEN MERGED GLOBAL MODEL IS INACCURATE?

We conduct a case study, where we record the performance of each client on its own local dataset after it receives the parameters from the global model. Higher performance at this moment indicates lower gradient norms according to Eq. (9).

In Fig. 9, one client (C4) consistently behaves notably worse than others in the baseline model. In fact it is common in baseline models and sometimes more than one clients behaves this way. And the global model is biased towards some classes (Fig .3). We observe this phenomenon happens to some of the clients in the following cases:

**Imbalanced classes** For instance, during the first task in EMNIST-L, Client 1 learns classes {18, 23}, Client 2 learns classes {10, 11}, Client 3 learns classes {15, 23}. In this case, after model averaging on the server, performance on Client 2's local data is impaired and is notably worse than others. Because Client 1 and Client 3 both learn from class 23, the merged model tends to be biased towards class 23.

**Different difficulty levels (a)** For instance, during the second task in EMNIST-L, Client 4 learns classes {14, 21}, none of which are learnt by other clients before. Client 1 learns classes {2, 20}, where class 2 is learnt by *Client 3* in *Client 3's* previous task. In this case, the merged model has worse performance on the Client 4's private task than on Client 1's. This is because the merged model tends to be biased towards easier classes.

**Different difficulty levels (b)** On EMNIST-B, We also observe that, even when classes are balanced and all incoming classes are not learnt by any of the clients, i.e., it is not one of the two mentioned cases above, some clients still learn worse than others. We attribute it to the differences of the inherit difficulty of different tasks. Clients that are learning more difficult tasks tend to have worse performance on their local data immediately after receiving the merged global model from the server.

### E.1.2 SOLUTIONS: HOW FEDCIL WORKS IN THE CASES?

In model consolidation, synthetic samples, which are used to consolidate the merged model, are generated by uploaded client models respectively *before* they are merged to a global model. Thus the generated image quality is not affected by three cases above. Therefore, consolidating the merged model with these high-quality synthetic images largely eliminates the model performance degradation caused by above cases. Besides, we generate balanced samples conditioned on classes, which additionally helps to solve the class imbalance problem.

After our model consolidation, the server model is less biased towards different clients' tasks or classes. Fig. 3 and Fig. 9 illustrate our effectiveness towards this aspect. Our consistency enforcement does the same thing, but from the perspective of client side: we distill the knowledge from the global model to local clients by generating class-balanced samples with the global model.

To summarize, in FedCIL, each client, which is initialized from the our better consolidated global model, makes more accurate and consistent predictions at the early stage of the local training. It largely prevents the ACGAN from the early gradient exploding problem.

### E.2 FEATURE NORMS

According to Eq. (9), lower feature norms also help to reduce the gradient norms. Previous study (Xu et al., 2020) reveals that knowledge distillation can help to reduce the overly high feature norms caused by noisy samples.

From this perspective, the consistency enforcement, which is also a kind of knowledge distillation in terms of form, can help to reduce the feature norm when training with noisy generated samples, in addition to its benefits on learning less biased models mentioned in Section 4.3 and appendix E.1.

## F DISCUSSIONS

### F.1 DIFFERENCES WITH REACGAN

From a high-level view, Both ReACGAN Kang et al. (2021) and our method can stabilize the training process of ACGANs, but they are designed to solve entirely different problems with different motivations. Thus it is not surprising that ReACGAN hardly helps to improve the model performance in Tab. 3.

| Model | Accuracy |
|---|---|
| FedAvg + DGR | 63.55 |
| FedAvg + ACGAN Replay | 66.19 |
| FedAvg + ReACGAN Replay | 65.87 |
| FedCIL (Ours) | **73.12** |

Table 3: Model Performance with Difference GR Strategies on EMNIST-B.

ReACGAN stabilizes the offline training of one ACGAN by exploiting feature normalization and relational information in the class-labeled dataset. It doesn't consider challenges from federated learning and continual learning. In other words, it is neither a federated learning nor a continual learning algorithm. Instead, FedCIL stabilizes the distributed training of multiple ACGANs by improving the server-side aggregation of client models and client-side feature learning (with the help of the global model). Our model consolidation and consistency enforcement are designed for the non-iid distribution and the catastrophic forgetting challenges in our continual federated learning setting. ReACGGN is not designed for these purposes.

## F.2 LOCAL PERFORMANCE

A typical federated learning scenario aim to learn a global model with distributed data, and thus local performance is usually not considered in evaluation. We added experiments to see how the private performance vary across different models. Tab 4 shows the results.

| Model | Accuracy |
|---|---|
| FedProx + DGR | 85.97 |
| FedProx + ACGAN Replay | 86.40 |
| FedCIL (Ours) | **88.73** |

Table 4: Local Model Performance on EMNIST-L.

It is interesting that our model has higher average private accuracy on the client side. We agree that in federated learning, better global accuracy often contradicts local accuracy. In the extreme case, if each client only does local training without communications, it is more likely to have higher performance on its own dataset with low global accuracy. Here we believe the continual learning part plays an important role. Because our model better remembers previous knowledge, the extra benefits from continual learning leads to its higher accuracy.

