# OpenReview forum: "Better Generative Replay for Continual Federated Learning"
_ICLR.cc/2023/Conference — ICLR 2023 poster_

### Official Review · Reviewer_qLZV · 2022-10-17

**Confidence:** 3
**Correctness:** 4
**Technical Novelty And Significance:** 2
**Empirical Novelty And Significance:** 2
**Recommendation:** 6

**Clarity, Quality, Novelty And Reproducibility:**

The core idea is clearly presented and the presented tricks are novel. The study has reasonably good quality.

**Strength And Weaknesses:**

The paper is well written and the topic is well motivated in the introduction. The contribution is clearly mentioned and explained.
The experimental results seem to outperform the state of the art approaches.

It mainly addresses the instability in this specific learning framework by enhancing the generalization of both clients and servers. Consequently, some locality should be sacrificed. Have you observed this in your experiments?


The proposed tricks seem to be effective and interesting however I doubt the significance of the contributions make it succeed to be published as a full paper at ICLR.

**Summary Of The Paper:**

In the continual federated learning, the stream input data usually is provided non__iid which affects adversely on the learning performance.  The authors used generative replay idea for continual learning, in which there is no need to transfer data among the clients and servers (in federated learning framework), rather the learned parameters can be shared. Namely, Auxiliary classifier GAN (ACGAN), a study from Odena et al. (2017) has been used as the generative model to avoid forgetting of the previously learned tasks. As the mentioned non_iid nature of the incoming data makes the federated learning difficult and using ACGAN as the generative replay fails, the authors proposed two technical tricks which could help this combination succeed.  The first trick, called as model consolidation, in which the clients parameters are aggregated and used to initialize the server parameter. Then, from the clients generators, balanced synthetic data is generated to use for enhancing training in the server side.
As the second trick, “consistency enforcement”, is performed in the client side. Each client’s loss includes two terms. In the first term, the new coming training data  are mixed with the synthetic data generated by the global generator conditioned on the labels from previous tasks, hence the client not only learns the new classes, but also retains the previously learned classes.
For the second term, consistency loss function, the idea is to include kullback divergence to enhance generating the same distribution for the real data and the data generated from the global model generator and the generated data of the past or current task generators, when the label is the same.

**Summary Of The Review:**

At this point, this paper is a border-line study in my opinion.

---

> ### Author Response · Authors · 2022-11-15
> **Response to Reviewer qLZV**
>
> Thank you very much for reviewing our paper and providing constructive comments!
>
> ### Q1 Local performance
>
> A typical federated learning scenario aim to learn a global model with distributed data, and thus local performance is usually not considered in evaluation [1,2,3].
>
> According to your suggestion, we added experiments to see the private performance (experiment on EMNIST-L).
>
> | Model                  | Accuracy  |
> | ---------------------- | --------- |
> | FedProx + DGR          | 85.97     |
> | FedProx + ACGAN Replay | 86.40     |
> | FedCIL (Ours)          | **88.73** |
>
> It is interesting that our model has higher average private accuracy on the client side. We agree that in federated learning, better global accuracy often contradicts local accuracy. In the extreme case, if each client only does local training without communications, it is more likely to have higher performance on its own dataset with low global accuracy.
>
> Here we believe the continual learning part plays an important role. Because our model better remembers previous knowledge, the extra benefits from continual learning leads to its higher accuracy.
>
> ### Q2 Contribution
>
> Thanks for recognizing the effectiveness of our model. We have two major contributions as summarized in the paper:
>
> **1. The first contribution**: We propose a new challenging and practical problem: the continual federated learning problem, which follows the standard paradigm of continual learning [4] and federated learning [1,2].
>
> **2. The second contribution**: Our method is also a new federated learning algorithm for generative models. Improving generative models in federated learning is also an under-explored problem yet. We are among the first to discover and research this problem.  In our paper diagnose (**Sec. 4.2** and **Appendix E**) and improve (**Sec. 4.3, Sec 5.4, Appendix E**) the training issues of generative models in the context of federated learning.
>
>
> With respect to the second contribution, we would appreciate it if you could take a look at our newly added Appenix E in our revised paper. It further illustrates how the proposed two modules help to correct the biases in models, which leads to lower gradient norms.
>
> With lower gradient norms, then as illustrated in Sec. 4.2, better results are produced by this "chain": lower gradient norm -> eq. (8) is stable -> lower FID -> better generative replay -> better results.
>
> We thank you again for your review and raising the interesting local performance question. We look forward to your reply and having more discussions with you if you have more questions!
>
> **Reference**
>
> [1] McMahan et al., Communication-efficient learning of deep networks from decentralized data
>
> [2] Zhu et al., Data-free knowledge distillation for heterogeneous federated learning. ICML 2021
>
> [3] Li et al., Federated optimization in heterogeneous networks, ICML 2020
>
> [4] Van de ven et al., Three scenarios for continual learning, 2019

---

> > ### Comment · Reviewer_qLZV · 2022-12-10
> > **Contribution**
> >
> > Thank you very much for the provided responses and new modifications. I think this study is interesting and seems to work. However I still believe this study is borderline. The first contribution of proposing a continual federated learning structure is not novel. The second contribution of using generative models is more convincing thought. The proposed model consolidation and consistency enforcement seems to be effective. So I update my vote for this study to marginally above the acceptance threshold.

---

> > > ### Author Response · Authors · 2022-12-10
> > > **Thanks for the reviewer response**
> > >
> > > Dear Reviewer qLZV,
> > >
> > > We sincerely thank you for your review and the recognition of our methods! We will include the discussions on the local performance in our final version.
> > >
> > > Sincerely,
> > > Authors of Paper 2381

---

> ### Author Response · Authors · 2022-12-01
> **Follow up**
>
> Thanks again for your constructive comments. Would you mind checking our responses to see if your previous concerns/questions have been addressed?
>
> We are also more than happy to provide additional experiments and discussions if you have further questions.

---

### Official Review · Reviewer_S7Qy · 2022-10-25

**Confidence:** 3
**Correctness:** 3
**Technical Novelty And Significance:** 3
**Empirical Novelty And Significance:** 3
**Recommendation:** 6

**Clarity, Quality, Novelty And Reproducibility:**

This paper is not written well and is hard to follow. However, the new problem setup and the method proposed in this paper are quite original.

**Strength And Weaknesses:**

### Strong points

- This paper introduces continual federated learning, which is a challenging but more realistic problem than existing federated learning. Continual learning without a memory buffer in the federated setting is an interesting and important problem
- The idea of finetuning the server model using individual clients’ generators (expressed as “model consolidation” in the paper) seems original to me. Since the server can receive the generators trained with individual local data distributions, fine-tuning the averaged server model with the synthetic data is a straightforward approach for transferring local knowledge to the server model.

### Weak points

- The experiments are difficult to understand, and missing details.
    - How many communications are performed for each task?? It seems that the degenerated server aggregation issue pointed out in the paper can be alleviated via enough communication per task.
    - How the classification nodes are added and aggregated when training the next task?
    - How many synthetic images are generated for each generator per iteration?
- Evaluation on the more challenging datasets is required such as CIFAR-100 or ImageNet
- Each client should transfer the generator to the server for every communication round, which can raise privacy concerns since we can easily synthesize the raw data using the generator.
- The writing is hard to follow, a few statements are vague, and notations are imprecise
    - Why do the proposed two methods mitigate the increased loss issue at server aggregation?
    - Subscript at $\theta$ denotes a client index in the left figure in Figure 1, but the subscript denotes task index in other parts of the paper, e.g., eq. 11
- Questions
    - $\theta_{\text{global}}$ in Eq. 10 only denotes the parameter of the generator? or denotes parameters of the generator, discriminator, and classifier?
- FedCIL requires more communication cost than FedAvg or FedProx + DGR since each client communicates all parameters of the model including generator, discriminator, and classifiers.

### Minor comments

1. Typos
    1. Page 5: federated leading → federated learning
2. The paper should include a discussion about the recent federated learning approach [1].


### Reference

[1] J. Dong et al., Federated Class-Incremental Learning, CVPR, 2022

**Summary Of The Paper:**

This paper introduces a continual federated learning problem where each client deals with a class-continual learning problem with a sequence of tasks independently, and the server learns the tasks of all clients via simple model averaging. This paper observes that the aggregate server model suffers high loss and FID score on individual client data, and proposes two methods (model consolidation and consistency enforcement) to handle this issue. Extensive experiments on multiple classification datasets show that their method has significant advantages over baseline methods which naively adopt data-free class-incremental learning approaches on a federated learning framework.

**Summary Of The Review:**

While the writing needs to be improved and a more comprehensive evaluation is required, I believe that this paper has two major contributions: 1) it proposes a realistic continual federated learning problem, 2) it tackles the problem of training a GAN-based model in federated learning in the presence of data heterogeneity over clients.

---

> ### Author Response · Authors · 2022-11-15
> **Response to Reviewer S7Qy**
>
> Thank you very much for your time and effort for this review. We appreciate your recognition and constructive questions about our work!
>
> ### Q1. Experimental Details
>
> Thank you again for your constructive comments on experiments. In the following we provide more details about our experiments, which have also been added to our revised paper.
>
> **Q1.1: round per task.** As mentioned in Sec. 5.3, we set local training iteration T=400 and global communication round R=200. There are five private tasks for a client, thus for a client, there are 200/5 = 40 communication rounds with the global model per task.
>
> In a typical Federated Learning setting with classification models, more communication rounds usually lead to better results [1,3]. Existing works [1,2,3] often set the communication round R to the same constant value for all compared models, in order to have fair comparisons (e.g., R=200 in [2]). In our experiments, we follow [2] and simply set the communication round to 200.
>
> We also show the results of different communication rounds as follows. The results show that more communication rounds lead to improvements for all models, while our method still outperforms baselines.
>
> | Number of R            | 100       | 200       | 300       | 400       |
> | ---------------------- | --------- | --------- | --------- | --------- |
> | FedProx + DGR          | 68.01     | 71.83     | 73.28     | 74.91     |
> | FedProx + ACGAN Replay | 69.48     | 73.91     | 75.74     | 76.82     |
> | **FedCIL (Ours)**      | **75.89** | **78.15** | **80.34** | **81.99** |
>
> **Q1.2: new nodes per task**
>
> We add two new nodes for new classes per task (details are available in Appendix A).
>
> **Q1.3: images generated per iter**
>
> It is the same as *mini batch size B*, which is 100 for CIFAR-10 and 32 for others.
>
> ### Q2. More Evaluation
>
> As suggested, we conduct an experiment on CIFAR-100. We build the dataset following the steps in Appendix A. FedCIL is still more effective over baselines.
>
> | Model                  | Accuracy  |
> | ---------------------- | --------- |
> | FedProx + DGR          | 12.21     |
> | FedProx + ACGAN Replay | 12.35     |
> | FedCIL (Ours)          | **14.57** |
>
> ### Q3. Privacy
>
> The privacy concerns are discussed in *Sec.4.3 Privacy*. Because developing more advanced private GANs is not the major focus of this paper, we meet a basic privacy level by parameter sharing, following [5,6]. If higher privacy levels are indeed required, existing works like [6] can be integrated into our model.
>
> ### Q4. Statements, Notations
>
> **Q4.1: How does two proposed modules work**
>
> Thanks for the constructive question! **We have added a new Appendix section, Appendix E** in the revised paper, which explains it in detail.
>
>  In Sec. 4.2, we demonstrate why baseline models fail. Quantitative results and Figures. 2-4 show that our two modules yield better results by stabilizing the training process. ( The logic is: lower gradient norm -> eq. (8) is stable -> lower FID -> better generative replay -> better results ).
>
> In **Appendix E**, we further explain how the two modules lead to lower gradient norms, in addition to  Sec. 4.3 and Sec. 5.4.
>
> **Q4.2: Notation**
>
> Thank you for pointing out this issue. We revised it in blue.
>
> ### Q5. θ_global in Eq.10
>
> It denotes the parameters of the generator and discriminator, which includes the classification head and discrimination head.
>
> In fact we tried to do consolidation with only discriminator or generator. The results are worse than doing consolidation with both. We think it is because the 'fix one and train another' paradigm breaks the balance of the two parts.
>
> ### Q5. Communication cost
>
> We agree that FedCIL requires more communication cost than baselines. It is a compromise. Compared to FedProx, FedCIL requires more communication cost, but effectively performs continual learning. Compared to FedProx+DGR, FedCIL has better performance and saves local space (Fig. 7 in Appendix B).
>
> ### Minor comments
>
> Thank you for pointing out the typo. We have corrected it in the revised paper. We have also discussed the referred work [7] in Sec. 2 in blue.
>
> Thank you again for your review and giving many constructive comments. We look forward to your reply and having more discussions with you if you have more questions or suggestions!
>
> **Reference**
>
> [1] McMahan et al., Communication-efficient learning of deep networks from decentralized data
>
> [2] Zhu et al., Data-free knowledge distillation for heterogeneous federated learning. ICML 2021
>
> [3] Li et al., Federated optimization in heterogeneous networks, ICML 2020
>
> [4] Xin et al., Private fl-gan: Differential privacy synthetic data generation based on federated learning, ICASSP 2020
>
> [5] Zhang et al., Training federated gans with theoretical guarantees: A universal aggregation approach, 2021
>
> [6] Rasouli et al., Fedgan: Federated generative adversarial networks for distributed data, 2020
>
> [7] Dong et al., Federated Class-Incremental Learning, CVPR, 2022

---

### Official Review · Reviewer_sMxb · 2022-11-05

**Confidence:** 2
**Correctness:** 3
**Technical Novelty And Significance:** 2
**Empirical Novelty And Significance:** 2
**Recommendation:** 5

**Clarity, Quality, Novelty And Reproducibility:**

The paper is written clearly. The work is not as novel as claimed. The code was not submitted so reproducibility could not be evaluated.

**Strength And Weaknesses:**

Strength
1. The work is reasonably well written and easy to follow.
2. The empirical results of the proposed approach (Table 1) is strong compared to baselines.

Weakness
1. Overclaiming. The authors frame “proposing the continual federated learning problem” as a major contribution, but much of the related work already focuses on various forms of continual federated learning despite slight difference from the proposed setting (e.g., class overlap).
2. Relevance. It felt to me that the authors curated a very narrow setting for which their method can outperform existing baselines (e.g., multi-task, global model, no class overlap, etc). If this curated setting is indeed highly relevant, it would be nice to motivate the proposed setting with practical examples.
3. It is unclear from the experimental results how crucial model consolidation and enforced consistency are in stabilizing ACGAN. Ablations or inclusions of other approaches to stabilization can help with this.


**Summary Of The Paper:**

This paper proposes the continual federated learning problem which has properties of both federated learning and continual learning, and proposes modifications such as model consolidation and enforced consistency to stabilize ACGAN to solve the proposed problem. The proposed approach achieves the best performance in the proposed setting, and achieves stable training similar to single-model learning of ACGAN.


**Summary Of The Review:**

Due to concerns around the relevance of the problem, the lack of practical motivation, and the overclaimed contribution, I recommend reject for the manuscript.

---

> ### Author Response · Authors · 2022-11-15
> **Response to Reviewer sMxb**
>
> We sincerely thank you for your time and effort for this review! In the following, we would like to clarify our (1). problem setting contribution, (2). relevance, and (3). ablation studies. We hope these explanations could address your concerns.
>
> ### Q1:
>
> We have two major contributions as summarized in the paper: we propose (1) a problem setting and (2) a federated learning algorithm for generative models (We are also among the first to investigate it).  We'd like to explain them.
>
> **1. Problem Setting**
>
> We agree that our problem setting feel similar with [1,6,8] at first look, but in fact is very different.
>
> Compared with existing works [1,6], our setting is the more general and standard continual federated learning setting. In particular, because we only apply two basic rules: (1). the standard definition of continual learning [3], i.e., **no class overlap** for a client’s learning sequence, and (2). standard federated learning: learn a **global model** with distributed data [4,5]. No further assumptions are added.
>
> To further justify the major differences, we explain them from two aspects, i.e., class overlap and global model.
>
> **(1) Class overlap.** ‘No class overlap’ or no ‘instance overlap’ [1] leads to two substantially different learning paradigms: Continual Learning (CL) [3] and Online Learning (OL) [2,7]. They are different problems with different challenges and solutions. The forgetting of learnt classes is the typical challenge in CL due to the ‘no class overlap’ rule [3].  Thus it is a critical difference, instead of a slight difference. We strictly follow the definition of CIL in [3] without any more assumptions, while FedLwF-2T [6] poses more assumptions on instance overlap and communication rounds (more discussion in Appendix D).
>
> **(2) Global model.** Typical federated learning [4,5] aims to learn a global model, which we follow. FedWeIT [8] is entirely different from us in that it *does not* learn a global model. It focuses on clients. Thus, it is a variant of continual learning model (with the help of other clients). While our FedCIL is a federated learning model.
>
> **2. Algorithm/Technical Contribution**
>
> We also propose a new federated learning algorithm for generative models. Improving generative models in federated learning is still an under-explored problem and we are among the first to investigate it. We diagnose (**Sec. 4.2** and **Appendix E**) and improve (**Sec. 4.3, Sec 5.4, Appendix E**) the training issues of generative models in the context of federated learning. This is also a major contribution of our paper.
>
> ### Q2:
>
> In fact, our problem setting is not a narrow setting, as explained above. Instead, we are the more standard continual federated learning setting with only two basic rules for continual learning [3] (no class overlap) and federated learning [1,4,5]. No other assumptions are added. Appendix D clarifies more details.
>
> ### Q3:
>
> (1) Actually, we have reported the ablation results in Table. 2. Both 2 modules help to improve the model performance notably.  FedProx [5] is a representative and more advanced FL algorithm than FedAvg [4] in distributed training. Table.1 shows it is not helpful here. Our methods are more effective.
>
> (2) In addition, Fig.2 and Fig.4 in our paper help to explain why results are improved: in particular, our methods are able to solve the unstable training problem mentioned in Sec. 4.2. The more stable training process leads to better model performance.
>
> (3) Fig. 3 explains that our method is more unbiased, which yields lower gradient norms according to Eq. 9. This leads to more stable training of ACGAN.
>
> (4) If you are further interested in how our methods stabilize the training in detail, you could take a look at **our newly added Appendix E: Case Study** in our revised paper. In short, our modules are effective in correcting biases in models, which leads to lower gradient norms. Then, *lower gradient norm* -> *eq. (8) is stable* -> *lower FID* -> *better generative replay* -> *better results* (as illustrated in Sec. 4.2)
>
> We look forward to your reply and having more discussions with you if you have any further questions! Thank you again for your review.
>
> **Reference**
>
> [1] Hendryx et al., Federated reconnaissance: Efficient, distributed, class-incremental learning. NeurIPS Workshop on NFFL, 2021.
>
> [2] Rosasco te al., Machine Learning: a Regularization Approach, MIT-9.520-Online Learning
>
> [3] Van de ven et al., Three scenarios for continual learning, 2019
>
> [4] McMahan et al., Communication-efficient learning of deep networks from decentralized data
>
> [5] Li et al., Federated optimization in heterogeneous networks, ICML 2020
>
> [6]Usmanova et al., A distillation-based approach integrating continual learning and federated learning for pervasive services. IJCAI Workshop on CMLIT, 2021.
>
> [7] https://en.wikipedia.org/wiki/Online_machine_learning
>
> [8] Yoon et al., Federated continual learning with weighted inter-client transfer. ICML 2021

---

> > ### Comment · Reviewer_sMxb · 2022-12-08
> > **Thanks for the author response**
> >
> > Thanks for providing the author response and the code, which I have read carefully. Despite the clarification regarding class overlap and global model, I still believe that "proposing the continual federated learning problem" as a major contribution is over-claiming given existing work (e.g., [1,6,8]). My expertise in continual learning and federated learning is limited, and therefore I defer the judgement of whether the "proposed" problem is novel and more general to other reviewers and the area chair.
> >
> > Given the newly added results, I do believe the proposed method is effective in solving the proposed problem. If the benefits of the proposed approach mostly come from stabilizing ACGAN, it would be interesting to see how other stabilization schemes such as ReACGAN perform for the proposed setting.
> >
> > Reference: Rebooting ACGAN: Auxiliary Classifier GANs with Stable Training

---

> > > ### Author Response · Authors · 2022-12-09
> > > **Thanks for the suggestions**
> > >
> > > Dear Reviewer sMxb,
> > >
> > > We sincerely thank you for your time and effort for this review! We experimented with the suggested ReACGAN [1] on the EMNIST-B dataset and also updated our uploaded code with the suggested ReACGAN ([updated code link](https://drive.google.com/file/d/1JzFxW0_xIufqbTrW-laHxL8GYYyy4-V5/view?usp=share_link)). We implemented it according to its [official implementation](https://github.com/POSTECH-CVLab/PyTorch-StudioGAN). We slightly modified the default hyperparameters to adapt to our case.
> > >
> > > ### Results
> > >
> > > | Model                       | Accuracy  |
> > > | --------------------------- | --------- |
> > > | FedAvg + DGR                | 63.55     |
> > > | FedAvg + ACGAN Replay       | 66.19     |
> > > | FedAvg + ReACGAN Replay [1] | 65.87     |
> > > | FedCIL (Ours)               | **73.12** |
> > >
> > > ### Analysis
> > >
> > > Results show that FedAvg + ReACGAN is not helpful in our setting. We think the results make sense, because ReACGAN and our FedCIL are designed to solve different problems with different motivations.
> > >
> > > * ReACGAN stabilizes the **offline** training of **one** ACGAN by exploiting feature normalization and relational information in the class-labeled dataset. It doesn't consider challenges from federated learning and continual learning. In other words, it is neither a federated learning nor a continual learning algorithm.
> > >
> > > * Our FedCIL stabilizes the **distributed training** of **multiple** ACGANs by improving the server-side aggregation of client models and client-side feature learning (with the help of the global model). Our model consolidation and consistency enforcement are designed for the *non-iid distribution* and the *catastrophic forgetting* challenges in our continual federated learning setting. ReACGGN is not designed for these purposes.
> > >
> > > Thank you again for your valuable suggestions. We will include the discussions on ReACGGN in our final version. We look forward to your reply and having more discussions with you if you have any further questions or suggestions!
> > >
> > > [1] Kang et al., Rebooting ACGAN: Auxiliary Classifier GANs with Stable Training, NIPS 2021
> > >
> > > Sincerely,
> > >
> > > Authors of Paper 2381

---

> > > ### Author Response · Authors · 2022-12-13
> > > **Follow up**
> > >
> > > Thanks again for your suggestions! We will make sure to clarify the contributions of our work in the final version to avoid potential over-claiming. In addition, would you mind letting us know whether our new results and analysis of ReACGAN can address your concerns or not? We look forward to your further feedback. Thanks!

---

> ### Author Response · Authors · 2022-11-18
> **A kind reminder of our released code**
>
> Thank you very much for your review! We have released our code with usage description:
>
> <https://drive.google.com/file/d/1VjePZuhyYXSrDgLARu3VGuCXO3_ZtgF1/view?usp=share_link>
>
> We look forward to your reply if you have any further questions or suggestions.

---

> ### Author Response · Authors · 2022-12-01
> **Follow up**
>
> Thanks again for your constructive comments. Would you mind checking our responses to see if your previous concerns/questions have been addressed?
>
> We are also more than happy to provide additional experiments and discussions if you have further questions.

---

### Author Response · Authors · 2022-11-18
**Source Code for FedCIL**

Dear Reviewers,

We shared the source code of our FedCIL with the usage description:
<https://drive.google.com/file/d/1VjePZuhyYXSrDgLARu3VGuCXO3_ZtgF1/view?usp=share_link>

---

### Author Response · Authors · 2022-12-09
**Summary of Author Responses and Paper Revision**

Dear Area Chairs and Reviewers,

Thank you very much for reviewing our paper and giving us many constructive comments, which are truly helpful in improving our paper.

We have provided detailed responses to your comments/questions. In the following, we would also like to briefly summarize the revisions made on our paper.

1. [Related Work] We added the discussion with (Dong et al., 2022) as suggested.
2. [Section 4.2&4.3] We corrected our notations.
3. [Section 5.4] We added more details on experimental configurations.
4. [Appendix E] We further explain how our proposed model consolidation and consistency enforcement improve the performance via case study from two perspectives (E.1 and E.2). For better illustration of E.1, we show cases where the merged global model becomes highly inaccurate (E.1.1) and explain why our method can solve it (E.1.2).

Thank you again for your insightful comments, and we look forward to your further feedback!

Sincerely,

Authors of Paper 2381

---

### Decision · Program_Chairs · 2023-01-20

**Decision:**

Accept: poster

**Justification For Why Not Higher Score:**

This is a work that considers continual learning for FL; the topic is interesting, however the authors could have done a better job in comparing with other SOTA methods and/or provide more evidence why this proposal works. However, the ideas presented worth to be published.

**Justification For Why Not Lower Score:**

The approach is novel and interesting by itself. The authors, through discussions, managed to flip some initially negative reviews, leading to an overall positive set of reviews.

**Metareview: Summary, Strengths And Weaknesses:**

Summary:

This paper proposes the continual federated learning problem which has properties of both federated learning and continual learning, and proposes modifications such as model consolidation and enforced consistency to stabilize ACGAN to solve the proposed problem. The proposed approach achieves the best performance in the proposed setting, and achieves stable training similar to single-model learning of ACGAN.

Strengths:

1. This paper studies the case of continual learning in federated learning, which is a challenging but more realistic problem than existing federated learning.
2. Continual learning without a memory buffer in the federated setting is an interesting and important problem
3. The use of GANs is interesting.

Weaknesses (initially):

1. Hard to understand experiments / incomplete experiments.
2. The authors could tone down the claim that this is the first work of applying continual learning in FL

Recommendation:

This is a paper that lies on the positive side of the review process. It is quite positive that authors have successfully addressed most (if not all) of the questions, suggesting changes in the final paper. We rely on the authors' discretion include all the proposed changes and improve the quality of the paper in its final version (especially for the concerns raised by the reviewers + incorporating the additional experiments conducted).

**Note From Pc:**

if the above contains the word "oral" or "spotlight" please see: "oral" presentation means -> notable-top-5% and "spotlight" means -> notable-top-25%. As stated in our emails, we are disassociating presentation type from AC recommendations